# An AI-Based Adaptive Surrogate Modeling Method for the In-Service Response of UVLED Modules

Cadmus Yuan 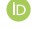

Department of Mechanical and Computer-Aided Engineering, Feng Chia University, Taichung 40724, Taiwan; cayuan@fcu.edu.tw; Tel.: +886-939-873-055

**Abstract:** The response forecasting of in-service complex electronic systems remains a challenge due to its uncertainty. An AI-based adaptive surrogate modeling method, including offline and online learning procedures, is proposed in this research for different systems with significant variety. The offline learning aims to abstract the knowledge from the known information and represent it as root models. The in-service response is modeled by a linear combination of the online learning of these root models against the continuous new measurement. This research applies a performance measurement dataset of the UVLED modules with considerable deviation to verify the proposed method. Part of the datasets is selected to generate the root models by offline learning, and these root models are applied to the online learning procedures for the adaptive surrogate model (ASM) of the different systems. The results show that after approximately 10 online learning iterations, the ASM achieves the capability of predicting 1000 h of response.

**Keywords:** AI-based adaptive surrogate model; response forecasting for in-service systems; uncertainty quantification; UVLED module; deep machine learning

## 1. Introduction

The complexity and dynamics of current electronic systems induce a high level of performance uncertainty during field application. Many of these complex systems consist of multiple components/subsystems, and the complexity induces multiple failure/deterioration mechanisms under multiple physical loadings, eventually increasing the performance uncertainties among the individuals [1,2]. An LED module is a good example of a complex system. It exhibits a variety of physics because it converts electrical energy to light and heat; it comprises multiple components, including the LED chip, wires, die-bonding material, and substrate. Considering the manufacturing uncertainties and different thermal degradation rates, the light output of the LED module might deviate from the statistical averages [3,4]. These biases increase in-service inconsistency and system maintenance costs.

Dragičević et al. [5] reviewed the recent reliability research regarding power electronics, identifying a paradigm shift toward the design-for-reliability (DfR) approach, and finding that the reliability performance of power devices is always investigated under different thermal loadings, because high temperature plays an important role in many relevant degradation mechanisms. Fan et al. [6] investigated the long-term reliability of LED packages, using thermal loading as one of the degradation factors. Yazdan et al. [7] studied the degradation of polymer materials, and correlated its impact to the light/color output of LED modules. Moreover, Lu et al. [8] further studied the color shift of LED modules. Sun et al. [9] studied the driver electronics in LED systems.

To facilitate the DfR concept, researchers have developed modeling techniques to predict the average reliability responses from a given set of design and loading parameters for complex electronic systems [10,11]. Conventionally, the physics-of-failure concept is always applied. By analyzing the failure/degradation mechanisms in detail, corresponding

physical-driven models have been developed for decades [12]. Due to the interactions between multiple failure/degradation mechanisms, AI-based reliability modeling methods have also been developed [4]. Zhao et al. [13] indicated that neural network methods are viable for power electronics because the significant development of computing hardware unleashes the potential of neural network methods in dealing with complex tasks, and the structure of neural networks is flexible enough for performance improvement.

Chou et al. [14] and Hsiao et al. [15] proposed deep machine learning modeling methods to replace the expert-driven finite element model. Yuan et al. [16] applied the long short-term memory (LSTM) method for the solder joint risk assessment of wafer-level chip-scale packaging (WLCSP) with limited datasets. Panigrahy et al. [17] overviewed the efficiencies and accuracies of the finite element method based on AI-assisted design-on-simulation methods, including artificial neural networks, recurrent neural networks, support-vector regression, kernel ridge regression, k-nearest neighbors, and random forests. Fan et al. [18] applied neural network architecture to model the spectral power distribution (SPD) of a light source. Yuan et al. [4] improved this method using a gated network with a two-step learning algorithm to build the empirical relationships between the design parameters, the thermal aging loading, and the SPD of LED products.

The maintenance costs of electronic systems grow as their complexity increases. Due to the uncertainties of the system, the maintenance cost is more than the material and operational costs, but the resource planning, storage, and management should also be included [19]. The accurate prediction of an in-service electronic system contributes to cost reduction of the system's maintenance cost. Jin et al. [20] applied a stochastic model to predict the failures of an in-service electric system by considering the latent failures. Grenyer et al. [21] reviewed the recent scientific approaches to the uncertainty engineering problem, and identified two major research gaps: the lack of frameworks to aggregate the multivariate uncertainty, and limited approaches to forecasting individual and aggregate uncertainty for complex engineering systems—especially for the in-service phase. Moreover, Grenyer et al. believe that deep learning techniques might contribute to the uncertainty forecast methods.

Li et al. [22] developed a structure-adjustable online learning neural network for gradually available data, such as in-service data, by applying an adjustable hidden layer to overcome the stability–plasticity dilemma of the online learning. Hu and Du [23] developed a time-dependent surrogate model with inner and outer loops. Moreover, Hu and Mahadevan [24] reported that the crossing points did not need to be highly accurate to predict the reliability during their study using the single-loop Kriging surrogate modeling method. Lieu et al. [25] developed an adaptive surrogate model based on a deep neural network by introducing a threshold to switch from a global prediction to a local one.

In this research, an AI-based adaptive surrogate modeling method that is able to represent and predict the individual product performance characteristics is established. As indicated in Figure 1, the hybrid machine learning method comprises offline learning and online learning parts. The experimental information is (partially) provided to the offline machine learning procedure in order to train the root models. Consequently, the same root models are input into the online machine learning to obtain the ASM against different training data. In this research, three sets of direct measurements of the UVLED module were applied to verify this algorithm. With careful definitions of error estimation, the function of offline learning, the predictive capability of the online learning procedure, and the ASM control scheme were analyzed.

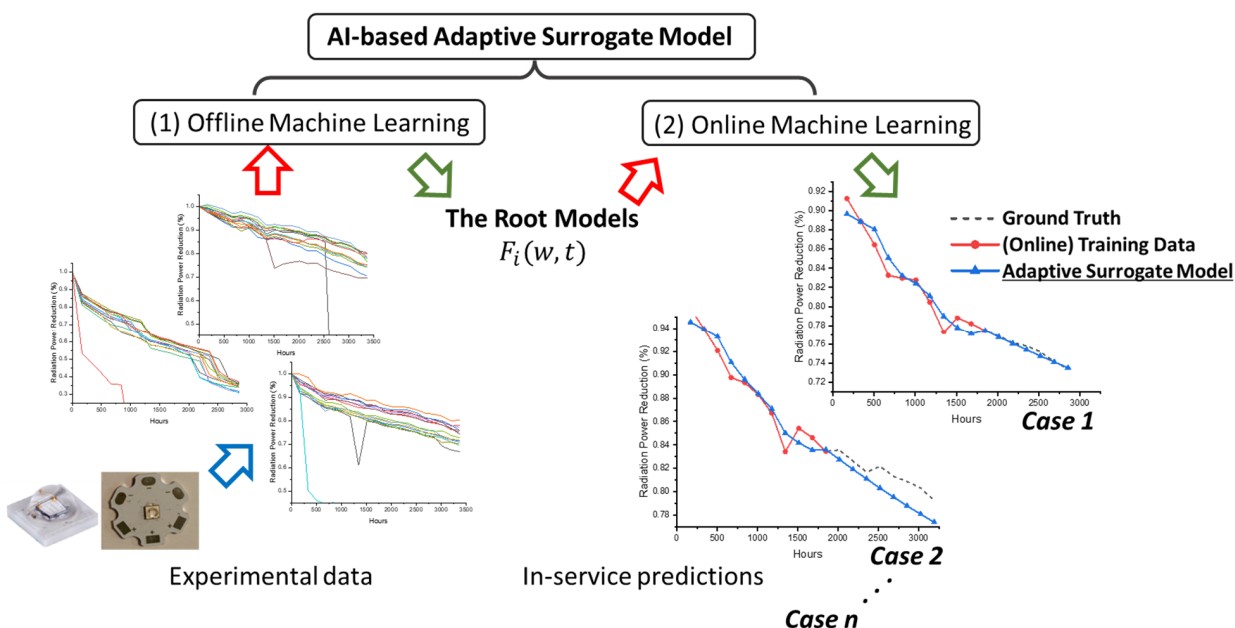

**Figure 1.** The architecture of the adaptive surrogate modeling method.

This paper is organized as follows: The fundamental scientific issues, field application requirements, and literature review are presented in the first section, "Introduction". The following section, "Methods", presents hybrid offline/online machine learning methods for adaptive surrogate models, along with the mathematical definition of the error estimations. The section "UVLED Measurement" describes the test object and the characteristics of the data. The section "Offline/Online Machine Learning" describes the implementation and results of hybrid machine learning. The following section, "Discussion", discusses the offline/online learning parameters and online learning control scheme to stabilize the learning against the measurement noise. The Conclusions of this paper are presented in the final section.

## 2. Methods

For an in-service electronic system, the response at time $t$ is written as $f(t)$. To emphasize the "future" response of such an in-service electronic system, we defined $t > t_m$, where $t_m$ is the current measurement time.

The $f(t)$ can be expressed as a linear combination of several evolved root models, as follows:

$$f(t) = \sum_i \alpha_{i,t_m} \cdot F_{i,\,t_m}(\vec{w}_{i,t_m}, \vec{p}(t)), \tag{1}$$

where $\alpha_{i,t_m}$ and $F_{i,tm}$ are the weightings and evolved root models that are obtained at time $t_m$, respectively. In this study, $F_{i,tm}$ represents a neural network, while $\vec{w}_{i,t_m}$ and $\vec{p}(t)$ represent the parameters obtained at time $t_m$ and the inputs at time $t$, respectively. This linear combination of evolved root models (Equation (1)) is defined as the adaptive surrogate model (ASM). It should be noted that the root model in this research is defined as a neural network model due to its numerical flexibility and information abstracting ability.

The hybrid machine learning consists of two steps, as illustrated in Figure 1. The first offline learning contributes to the $F_{i,0}$, and the collection of $F_{i,0}$ ($i = 1 \ldots n$) corresponds to the root models. By learning the measurement points at $t_m$, the weightings $\alpha_{i,t_m}$ and the evolved root models $F_{i,t_m}$ are achieved by the subsequent online learning procedure. It should be noted that one set of the root models can, in principle, be the start of several online learning procedures, as illustrated in Figure 1.

### 2.1. The Offline Machine Learning

The main purpose of the offline machine learning is to obtain stable and reliable root models. The experimental measurements are categorized into groups to form the training datasets for the offline machine learning, as $\mathcal{D}_i = \left\{ \left( \vec{p}_i, \vec{q}_i \right)_{t=0}, \left( \vec{p}_i, \vec{q}_i \right)_{t=1}, \dots \right\}$, where $\vec{p}_i$ and $\vec{q}_i$ are the input and output vectors, respectively. The backpropagation-based approaches with optimizers are taken to obtain the optimized weightings against the inputs and time, and form the *i*-th root models $F_i\left( \vec{w}_i, \vec{p}_i \right) = \vec{q}_i$.

To reduce the instability of the weightings of the neural-network-based approach, the combination of a genetic algorithm (GA) and the principal component analysis (PCA) framework, as described by Yuan et al. [26], was implemented. Moreover, the progressing GA optimizer and exponential kernel function for PCA can be expressed as follows:

$$K\left(x, x'\right) = \sigma \cdot \exp\left( -\frac{\| \, x - x' \, \|^2}{2} \right), \tag{2}$$

where $x$ and $x'$ are the genes obtained by different GA procedures, and the parameter $\sigma$ is set to 1.

### 2.2. The Online Machine Learning

The goal of the online machine learning is to obtain the stable weighting $\alpha_{i,t_m}$ and the evolved root models $F_{i,t_m}$. The in-service response $f(t)$ can then be expressed by the linear combination using Equation (1). Since the information obtained at time $t_m$ is prescribed by the root models, the stability–plasticity dilemma mentioned by Li et al. [22] can be reduced without complicating the neural network structure.

The online training set at the time $t_m$ is defined as $\mathcal{D}_{t_m} = \left\{ \vec{q}_{t_m}, \vec{q}_{t_{m-1}}, \dots, \vec{q}_{t_{m-n}} \right\}$, where $\vec{q}$ is the measured response vector, $n > 0$, and $m - n \geq 0$. In this research, balancing between using a large $n$ to avoid measurement instability and a small $n$ to increase the online machine learning speed, $n$ was set to 3.

The ASM weightings—$\alpha_{i,t_m}$ in Equation (1)—are defined as the ratio of the inverse of the online training errors (against $\mathcal{D}_{t_m}$). In each online machine learning step, in the parameters of the neural network $F_{i,t_m}$, $\vec{w}_{i,t_m}$ is updated by the machine learning process against $\mathcal{D}_{t_m}$. Learning whether $\vec{w}_{i,t_m}$ will remain at $t_{m+1}$ is an option for the online machine learning procedure.

To stabilize the online machine learning, the weighing change ratio is defined as follows:

$$WCR\left( \vec{w}, \vec{w}_0 \right) = \sqrt{ \frac{\sum_{j=1}^n \left( \frac{w_j - w_{j0}}{w_{j0}} \right)^2}{n} }, \tag{3}$$

where $\vec{w}_0$ and $\vec{w}$ are the neural network weighting vectors before and after the online machine learning, respectively, while $w_{j0}$ and $w_j$ are the components of the neural network weighting vectors, and $n$ is the component count of the vectors.

### 2.3. The Definitions of the Predictive Capability

Given a prediction accuracy tolerance $l_c$ ($l_c > 0$), and considering an in-service system at time $t_m$, the system response before $t_m$ is recorded by the hybrid machine learning algorithm into the ASM, but the information after $t_m$ is unknown. The predictive capability (PC) at $t_m$ is defined in two ways: First, from the practical point of view, the PC is defined by $PC_P$, and is an estimation based on the information obtained before $t_m$. Define $l_p$ as the error estimation of the ASM at time $t_p$, as $l_p = \| \, f\left(t_p\right) - \sum_i \alpha_{i,t_m} \cdot F_{i, \, t_m}\left(\vec{w}_{i,t_m}, \vec{p}\left(t_p\right)\right) \|_2$. Define the set $s_\alpha$ as a collection of $l_p$, which satisfies $l_p < l_c$ for $t_p \leq t_m$, but $\forall \beta - i > 0$, $l_{\beta - i} \notin s_\alpha$

if $l_\beta \geq l_c$. Define the function $C(\cdot)$ as a component counter of a set, and $PC_p$ is defined as follows:

$$PC_p = (C(s_\alpha) - 1) \cdot \Delta t, \tag{4}$$

where $\Delta t$ is the data measurement period.

On the other hand, to fine-tune the parameters for the hybrid machine learning procedure, one may apply known in-service historical responses. We define the prediction capability as PC from the oracle's point of view ($PC_O$). The term "oracle" comes from Greek mythology, and refers to someone able to communicate directly with the gods and give a response or message from the gods to someone else. In the actual application, the prediction at time $t + n$ ($n \in \mathbb{N}$) can be achieved at time $t$ by the ASM model, but not the prediction accuracy. This is because the actual system reaction at time $t + n$ is not available. However, in this parameter-tuning stage, the $t + n$ response can be known by using the historical responses. Therefore, the term "oracle" is applied to obtain a clear distinction between $PC_o$ and $PC_p$. Define the actual response as $g(t)$ and $l_t$ as the error estimation of the ASM at time $t$, which becomes

$$l_t = \| g(t) - \sum_i \alpha_{i,t_m} \cdot F_{i,\ t_m}(\vec{w}_{i,t_m}, \vec{p}(t)) \|_2. \tag{5}$$

The $PC_o$ can be defined as follows:

$$PC_o = \max t \text{ subject to } l_t, l_{t-1}, \dots l_{t-\gamma} \leq l_c \text{ and } t - \gamma > t_m,\ \gamma \geq 0 \tag{6}$$

From the application point of view, $PC_p$ should be smaller than and as close to $PC_o$ as possible, so as to ensure that the prediction is accurate and conservative. It depends on the characteristics of $g(t)$ and the online machine learning settings to stabilize the neural network training against $\mathcal{D}_{t_m}$ and the selection of $l_c$. The selection of $l_c$ is recommended to refer to the learning performance of the root models against $\mathcal{D}_i$.

## 3. UVLED Measurement

The UVLED packages (left panel of Figure 2a) were first mounted onto the metal-core printed circuit board (right panel of Figure 2a). The peak emission wavelength of the test samples ranged from 365 nm to 375 nm, with a rated driving current of 350 mA. Both the constant-stress acceleration degradation and step-stress acceleration degradation tests were implemented by Liang et al. [27]. Three experiments, with the input currents within the UVLED design range, were selected for this investigation, and are noted as sets A, B, and C. Table 1 lists the loading conditions of each test set. There were 14 samples in each set and the reliability measurement results, in terms of the radiation power reduction (in %) over time, are plotted in Figure 2b–d.

Analyzing the measurement data from Figure 2b–d, we can first eliminate the statistical outliers, such as A19 of Figure 2b, B1 and B9 of Figure 2b, and C46 and C52 of Figure 2c. The deviation of sets A, B, and C is plotted against time in Figure 3a–c, respectively. Deviation existed regardless of the constant-stress (set A) or the step-stress tests (sets B and C). The average deviation (i.e., the difference between the maximum and the minimum) over time was 0.0806, 0.0611, and 0.0860 for sets A, B, and C. Since the degradation tests were carried out within a controlled oven and power supplier, and the light output of the UVLED was measured using a calibrated integrated sphere [27], these deviations indicate that the in-service response of the UVLED system is influenced by the interaction of multiple causes of degradation.

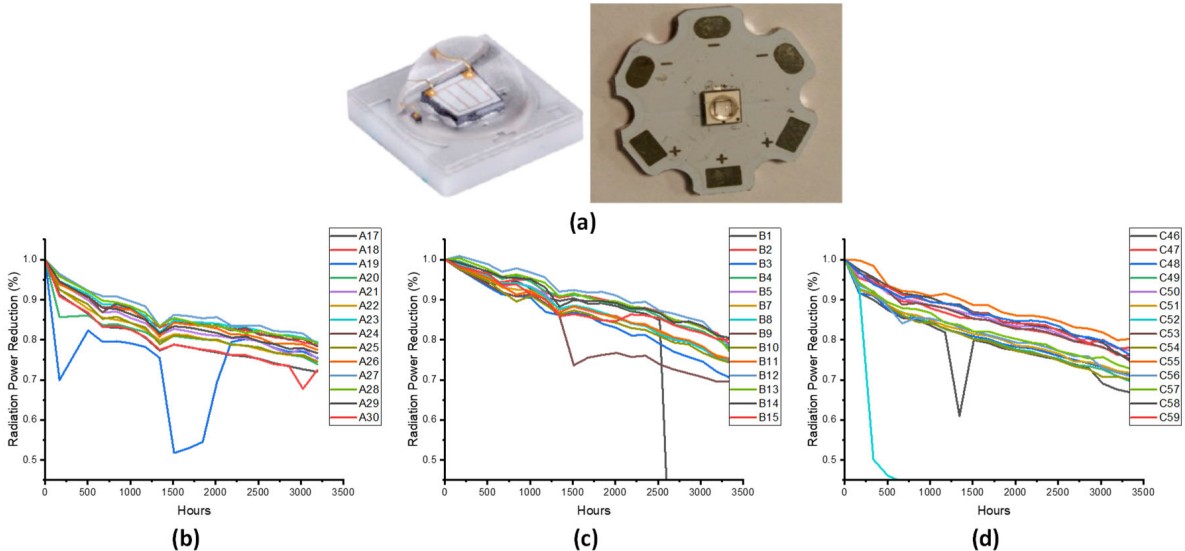

**Figure 2.** The UVLED reliability experiment and results. (**a**) is the sample of the sample, (**b**–**d**) are the radiation power reduction with respect to the Set A, B and C, respectively. The test conditions of Set A, B and C are listed in Table 1.

**Table 1.** The loading conditions of the UVLED reliability experiments.

| | Sets | | |
|---|---|---|---|
| | **A** | **B** | **C** |
| Samples | 14 | 14 | 14 |
| Temperature (°C) | 35 | 55 | 55–85 * |
| Current (mA) | 350 | 350–450 ** | 350 |
| Time (hours) | | Measured every 168 h | |

*: Base temperature started from 55 °C, and continuously increased every 504 h at a step size of 5 °C until it reached 85 °C. **: Input current started from 350 mA, and continuously increased every 504 h at a step size of 50 mA until it reached 450 mA.

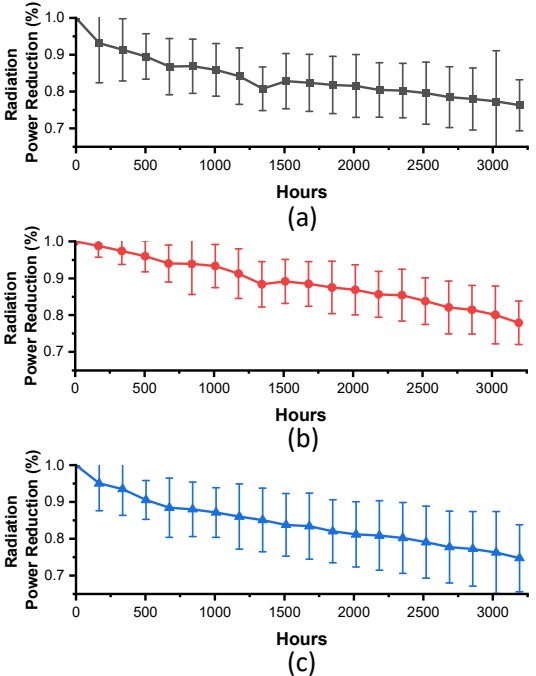

**Figure 3.** The deviation of the UVLED reliability test results: (**a**) set A; (**b**) set B; (**c**) set C.

## 4. Offline/Online Machine Learning

### 4.1. The Offline Machine Learning

The measurement data of sets B and C were categorized into four groups to increase their internal numerical similarity. Four offline training datasets were then formulated, as listed in Table 2. Each training set comprised one data group from set B and one from set C to increase the balance. Referring to Table 1, if only set B is selected for offline learning, the corresponding root model exhibits very limited ability to vary the base temperature. The same is true for only selecting set C.

**Table 2.** The offline training datasets and the learning results of the root models.

| Root Model | Training Sets | | Training Results | |
|:---:|:---:|:---:|:---:|:---:|
| | **B-Related** | **C-Related** | **B-Related Averaged Errors** | **C-Related Averaged Errors** |
| RM1 | b1 [1] | c1 [3] | 0.0145 | 0.0147 |
| RM2 | b2 [2] | c2 [4] | 0.0147 | 0.0272 |
| RM3 | b1 [1] | c2 [4] | 0.0154 | 0.0267 |
| RM4 | b2 [2] | c1 [3] | 0.0142 | 0.0146 |

[1]: Group b1 comprises cases B1, B2, B4, B13, B12, and B14. [2]: Group b2 comprises cases B3, B5, B7, B8, B10, and B11. [3]: Group c1 comprises cases C47, C50, C53, C55, C58, and C59. [4]: Group c2 comprises cases C46, C48, C49, C54, C56, C57.

At the initial stage of the offline model, the same structure was applied to the four root models. The "3,3,3,1" structure was applied, which has three inputs (i.e., base temperature, input current, and time), one output (the radiation power reduction), and two hidden layers (each with three neurons). The activation function was fixed to sigmoid.

Using the genetic algorithm proposed by Yuan et al. [26], 2000 initial neural network parameter sets were randomly generated for each GA run with the "progressing" GA optimizer, and each component of the chromosomes followed a zero-mean Gaussian distribution. Each chromosome was trained against the offline training datasets, and the best error norm was defined as the fitness ranking factor. The fitness ranking ensures that only the best five chromosomes can enter the next population. The iteration is converged when the norms of the new generation of chromosome vectors are less than 0.1. Moreover, a PCA was applied to the GA optimization results to obtain the PCA gene. Four GA runs were accomplished, resulting in the four best chromosomes for each offline training dataset. The super chromosome was then obtained by inputting these four chromosomes into the principal component analysis with the kernel function shown in Equation (2). To achieve the root models, an extra 10,000 backpropagation iterations (with a learning rate of 0.3) were applied to these four super chromosomes. Applying the error estimation given in Equation (5), the averaged errors of the data related to sets B and C are listed in Table 2, with the plots in Figure 4a–d. It should be noted that the "3,3,3,1" structure was chosen because it is the simplest structure with which the B- and C-related averaged errors are less than 0.03.

### 4.2. The Online Machine Learning

During the online machine learning, the dataset size was limited to three. The A17 measurement case within set A was first selected as the test case. Figure 5 shows the online machine learning procedure.

Figure 5a shows the initial stage, where the ASM curve (blue line) shows a significant difference from the dashed line (the ground truth). Figure 5b shows the first online learning, where three measurement points are collected for the root model training, and the ASM shows a change compared to Figure 5a. Figure 5c–j show the continuous online training processes, and one can observe that the ASM gradually comes closer to the ground truth.

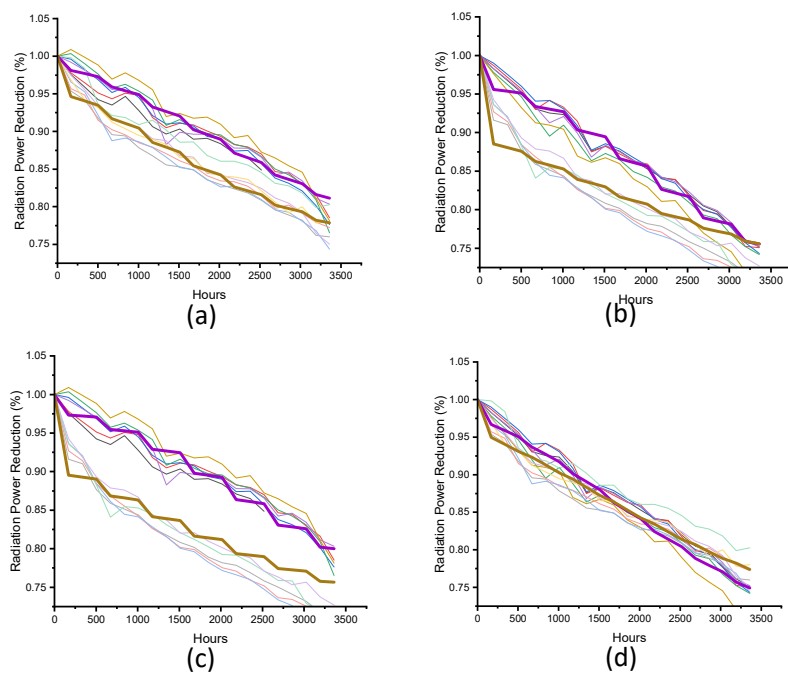

**Figure 4.** The training results of the root models (RMs): (**a**) RM1; (**b**) RM2; (**c**) RM3; (**d**) RM4. The dashed lines represent the measurement data listed in Table 2. The thick purple and dark yellow curves of each panel show the root model using the loading conditions set by sets B and C, respectively.

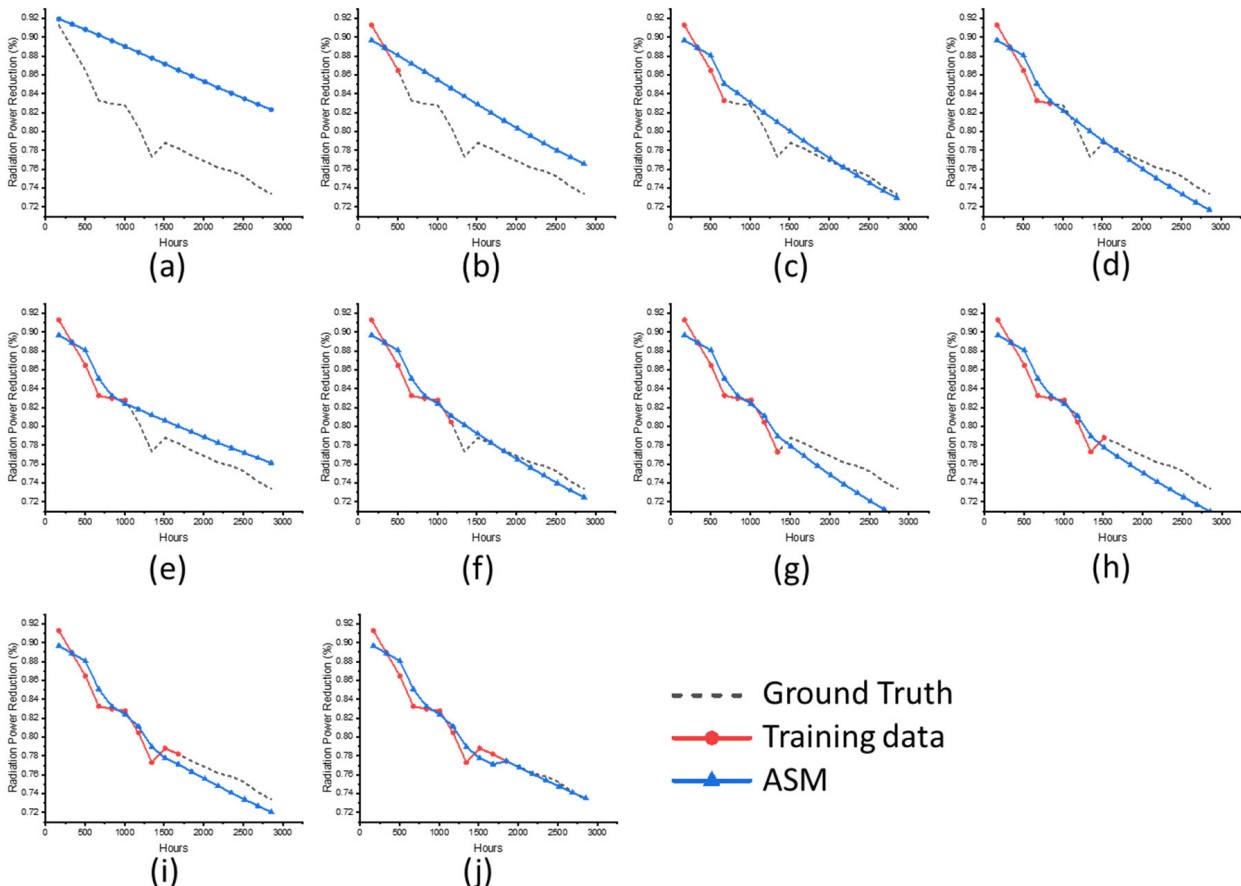

**Figure 5.** The online learning against case A17: (**a**) the initial state; (**b**–**j**) the 9 online learning steps.

Considering the quality of input data, four online learning points are worth mentioning: the radiation power reductions at 672, 1008, 1344, and 1512 h show significant discontinuity, and Figure 5c,e,g,h show the online learning processes, respectively. Although the discontinuous points worsen the ASMs' slope and their predictive capability, as depicted in Figure 5e,h, no significant divergence nor catastrophic failure of the ASM training was detected. This is mainly because there are multiple measurement points in the training dataset. The online machine training stopped at 1848 h due to the 11th ASM being very close to the ground truth, as shown in Figure 5j.

## 5. Discussion

### 5.1. The Quality of the Data

Beyond case A17, Figure 6 shows the hybrid learning results of cases A17, A25, and A28. From the helicopter's point of view, although the measurement data show significant differences (the dotted lines), with an average difference of 0.0635% at each time, all three ASMs (the solid lines) provide good predictive capabilities after a certain amount of online training. It should be noted that all ASMs in Figure 6 started with the same root models, shown in Figure 4.

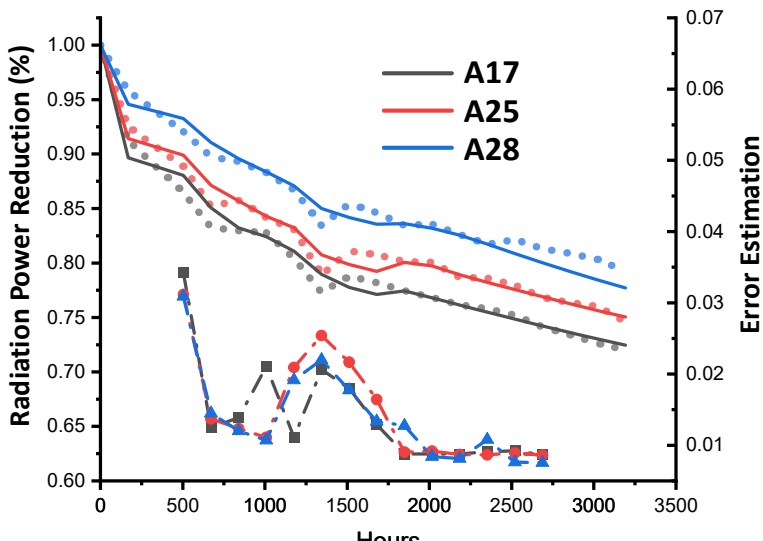

**Figure 6.** The online machine learning results of A17, A25, and A28. The upper curves show the ground truth (dotted line) and the best ASM, with the *Y*-axis on the left-hand side. The lower dashed curves with symbols (grey-square, red circle and blue-triangle) represent the error estimations of A17, A25 and A28, respectively. These errors are computed by Equation (5), with the *Y*-axis on the right-hand side.

The error estimations of the ASMs were obtained by Equation (5) from the oracle's point of view. Considering the ASM obtained at each $t_m$, each error estimation is the average of the product of Equation (5) throughout the whole time scale. Analyzing the changing trend of the error estimation, the error was high at the initial state and decreased after learning. However, the error increased at approximately 1344 h, and decreased again until a relatively low error plateau was reached. To understand this mechanism, one should refer to Figure 5g, which shows the ASM at the same time slot. When the measured data show a significant discontinuity, the data quality might impact the slope of the ASM, worsening its predictive capability. A similar condition occurs in cases A25 and A28 as well.

From the practical point of view, when the online learning reaches 1344 h, one should not avoid or ignore this data discontinuity, because the future ($t > 1344$ h) is unknown. However, when the time reaches the next measurement point—i.e., $t = 1344 + 168$ h—the previous discontinuity becomes the historical information, and is less interesting to the

online learning. Hence, this paper applies only the algorithm of multiple online training sets to stabilize the online training quality.

### 5.2. The Use of Predictive Capability ($PC_p$ and $PC_o$)

Using Equations (5) and (6) in the prediction of the in-service electronic system is ideal, but is not applicable, because the in-service response is unknown (until it reaches the measurement point) and comes with certain uncertainties. Another practical prediction capability estimator should be defined.

Figure 7a shows the online machine learning of case A17 at $t = 840$ h. The thick blue curve is the third ASM, and is written as ASM (3). The red dotted curve and the grey dashed curve are derived from ASM (3), which is known at $t = 840$. The light-grey dotted curve shows the ground truth, and the data after $t = 840$ h are unknown from the practical point of view. Hence, from the practical point of view, $PC_p$ is required to identify the predictive capability of ASM (3), shown as the red dotted curve in Figure 7a. A similar situation applies at $t = 1848$, as shown in Figure 7b.

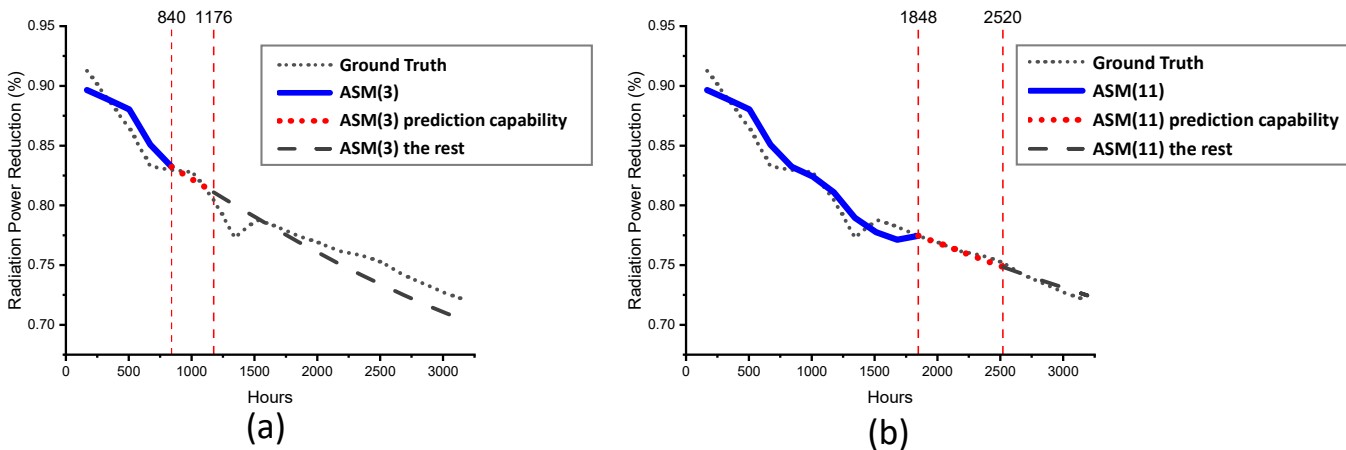

**Figure 7.** The illustration of the prediction capability concept: The hybrid machine learning of case A17 is applied. (**a**) and (**b**) are at $t = 840$ and 1848 h, respectively.

Due to the unavailability of the future data, the computation of $PC_p$ is limited to the information acquired at and before the time at which the ASM is obtained. First, $l_c$ was defined as 0.02 by considering the learning capabilities of the root models, as shown in Table 2. Then, Equation (4) was applied to compute the $PC_p$ by the quality of ASM.

Applying Equations (4) and (6), Table 3 was generated against cases A17, A25, and A28. The value of $PC_o - PC_p$ represents the quality of $PC_p$. If the value is positive, $PC_p$ is conservative; otherwise, $PC_p$ is aggressive. Analyzing $PC_o - PC_p$ in Table 3, one can observe that the negative values occur approximately between $t =$1008 and 1176 h, which is near the discontinuity point of 1344. Such discontinuity cannot be foreseen before the measurement point. The prediction is conservative after approximately 1680–1848 h due to the ground truth showing few fluctuations. Throughout the online machine learning process, the average difference is approximately lower than 2, meaning that the index of $PC_p$ is reasonable and conservative.

### 5.3. The Contribution of the Root Models

Referring to Equation (1), the weightings of the evolved root models and the changes in $\alpha_{i,t_m}$ at each online learning process represent the contribution of the evolved root models. The weightings during the online learning in cases A17, A25, and A28 are plotted in Figure 8. The weightings at the beginning of the online learning show certain similarities. However, the weightings evolved diversely along with the learning. All of the (evolved) root models showed the potential to be the one with the highest weighting at all learning stages. This also shows the equal importance of the four root models in this research.

**Table 3.** Comparison of predictive capabilities from the practical point of view and the oracle's point of view for cases A17, A25, and A28.

| Aging Time (Hours) | A17 | | | A25 | | | A28 | | |
|---|---|---|---|---|---|---|---|---|---|
| | $PC_p{}^*$ | $PC_o{}^*$ | $PC_0-PC_p{}^*$ | $PC_p{}^*$ | $PC_o{}^*$ | $PC_0-PC_p{}^*$ | $PC_p{}^*$ | $PC_o{}^*$ | $PC_0-PC_p{}^*$ |
| 504 | 1 | 1 | 0 | 1 | 1 | 0 | 1 | 1 | 0 |
| 672 | 2 | 4 | 2 | 2 | 4 | 2 | 2 | 4 | 2 |
| 840 | 2 | 5 | 3 | 2 | 5 | 3 | 2 | 5 | 3 |
| 1008 | 4 | 2 | −2 | 4 | 4 | 0 | 4 | 4 | 0 |
| 1176 | 4 | 3 | −1 | 4 | 1 | −3 | 4 | 1 | −3 |
| 1344 | 1 | 4 | 3 | 1 | 2 | 1 | 1 | 4 | 3 |
| 1512 | 2 | 4 | 2 | 2 | 2 | 0 | 2 | 6 | 4 |
| 1680 | 3 | 10 | 7 | 2 | 2 | 0 | 3 | 5 | 2 |
| 1848 | 4 | 9 | 5 | 2 | 9 | 7 | 4 | 6 | 2 |
| 2016 | 4 | 8 | 4 | 2 | 8 | 6 | 4 | 8 | 4 |
| 2184 | 4 | 7 | 3 | 3 | 7 | 4 | 5 | 7 | 2 |
| (Avg) | | | 1.93 | | | 1.71 | | | 1.50 |

\*: With the unit of $\Delta t = 168$ h.

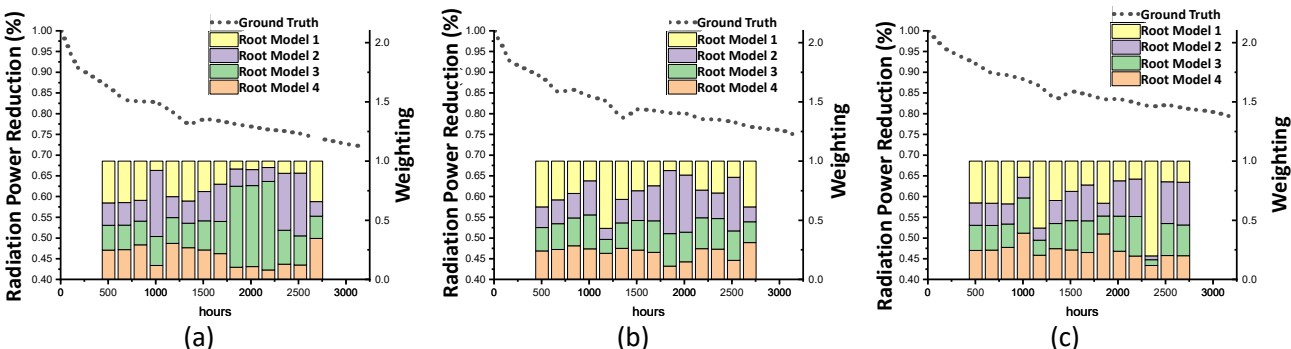

**Figure 8.** The contribution of the root models during the online machine learning: (**a**) A17; (**b**) A25; (**c**) A28. The dotted curves in panels (**a**–**c**) are the ground truth, with the Y-axis on the left-hand side. The bottom stacked column plots are the weightings of the evolved root models after each online learning iteration, and the corresponding Y-axis is located on the right-hand side.

*5.4. The Neural Network Weighting Control Scheme*

Each online machine learning procedure involves a backpropagation process that adjusts the weighting of the evolved root models. There are at least two starting stages for online learning: In the first, the backpropagation uses the weightings of the zero-hour root models as the initial weightings. After a considerable number of machine learning iterations, a low error can be achieved. This is called the "back-to-original" approach. On the other hand, the backpropagation can be triggered using the previous learning results and weightings as the starting point. This is called the "progressing" approach.

Both the "back-to-original" and "progressing" approaches were carried out against case B15 (Figure 9a), which was not used in any of the training sets of the root models. The same root models were prepared for the online machine training. The contributions of the root models in terms of $\alpha_{i,t_m}$ in Equation (1), for the "back-to-original" and "progressing" approaches, are plotted in Figure 9b,c, respectively. Compared to the differences between Figure 8a–c, a certain similarity can be found between Figure 9b,c. Moreover, the average difference in the error $l_t$ (Equation (4)) for all ASMs is approximately 0.008, and it is significantly lower than the errors in Table 2; therefore, one can conclude that the ASMs obtained from the "back-to-original" and "progressing" approaches are similar.

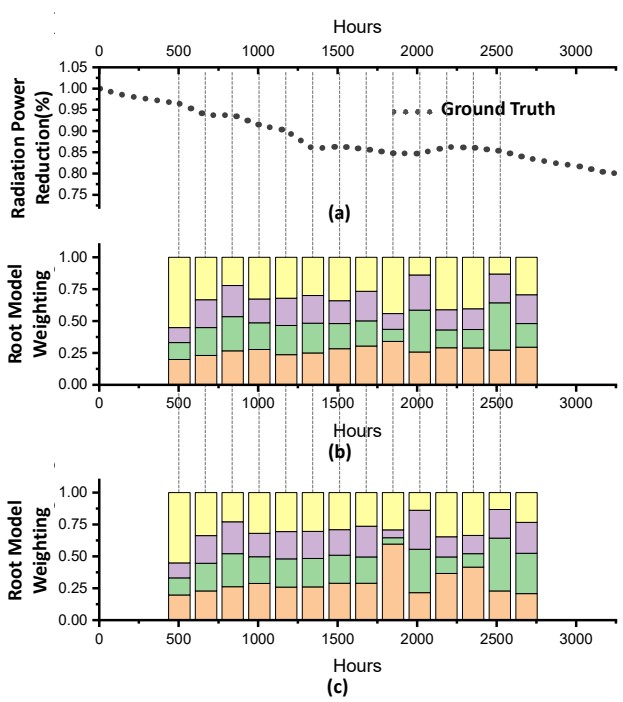

**Figure 9.** The contribution of root models for (**a**) case B15 using the (**b**) "back-to-original" and (**c**) "progressing" online machine learning optimizers.

However, the difference between the "back-to-original" and "progressing" approaches can be detected in the weighting change (based on Equation (3)) in the neural network. It is reasonable that the weighting change of the "progressing" approaches should be small, because the training sets are similar between any two online training steps next to one another. The weighting changes of the four root models were computed by Equation (3), as shown in Figure 10. Figure 10a shows that the weighting change of RM1 under the "progressing" approach is mostly small, but reaches its peak at approximately 2184 h, corresponding to the measurement data discontinuity point. The weighting changes of RM2 are small (Figure 10b), as is its contribution (Figure 9b,c). The weighting change of RM3 shows no significant change under the "back-to-original" approach, but not under the "progressing" approach. The weighting change of RM4 captures a significant peak at 2016 h. Hence, the "progressing" approach shows a low weighting change if the measured data are continuous, and it becomes sensitive when discontinuity of measurement is detected.

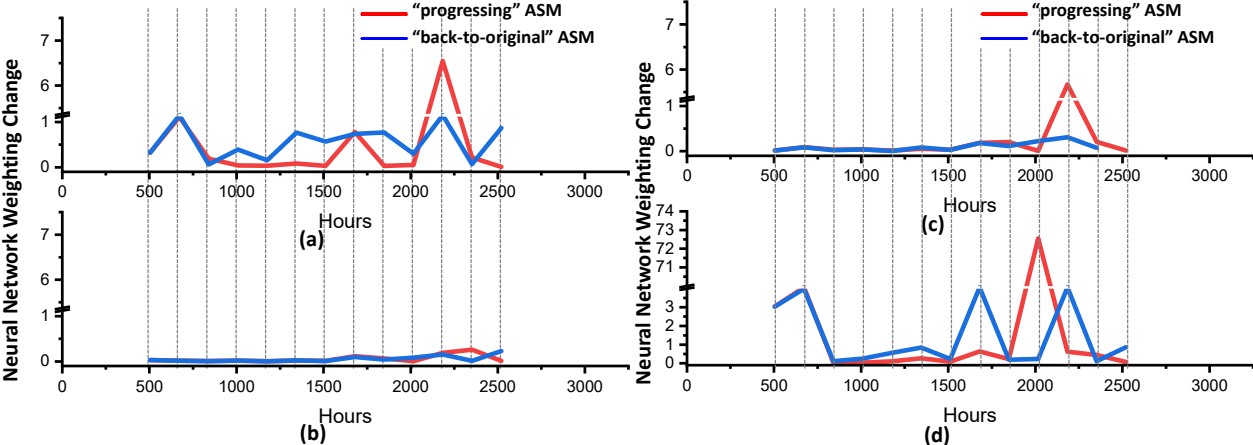

**Figure 10.** The neural network weighting change of root models during the online learning procedure when the "back-to-original" and "progressing" optimizers are applied: (**a**) RM1; (**b**) RM2; (**c**) RM3; (**d**) RM4.

## 6. Conclusions

In this research, an AI-based hybrid machine learning method, including offline and online machine learning, was developed to obtain an adaptive surrogate model (ASM) for the performance prediction of an in-service complex electronic system. The offline machine learning aims to obtain the root models (i.e., neural network models) based on known experience. Since the root models' quality impacts the later online learning performance, a genetic algorithm is recommended to obtain a stable root model. The ASM is available through the online learning algorithm against the available measurements, via a liner combination based on Equation (1).

Three sets of UVLED module performance measurements were used for the validation of the hybrid machine learning, with three input parameters, including the case temperature, input current, and time, as well as the output of the radiation power reduction (in %). Via the offline learning, including the genetic algorithm and principle component analysis processes, four root models were obtained, with an error norm of 0.0179, and these four root models were applied for all of the online machine learning.

During the online machine learning, three data points are provided to the backpropagation to generate the ASM at each measurement point. The results show that when the measurement data exhibit significant discontinuity, the error of the ASM increases. These errors can be decreased after a few more online learning iterations.

Considering the unavailability of future measurement at the present time, the predictive capabilities from the practical point of view are defined by Equation (4). By defining the error criterion of 0.02, the average difference between the actual capability and the prediction is approximately $2 \cdot \Delta t$ ($\Delta t$ = 168 h), and the definition of the practical predictive capabilities is believed to be reasonable and conservative.

The contribution of the four root models was studied, and their equal importance was detected. To stabilize the online learning, the "back-to-original" and "progressing" approaches were investigated for their stability and sensitivity. The ASMs from both approaches performed with high similarity. However, the weighting change of the evolved root model was stable, and was very sensitive to the changes in the incoming data in the progressing approach. The quality of the root model influences the online learning performance, and more advanced genetic algorithms should be implemented, such as non-dominated sorting genetic algorithms (NSGAs).

In this research, the same four root models were applied to all online learning optimizations. When the loading condition ranges were within the root models, the ASM approached the real response after approximately 9–10 learning iterations (approximately 1500–1800 h), with the real prediction capability of more than $6 \cdot \Delta t$ ($\Delta t$ = 168 h), considering an average response deviation of 0.0760 and a given accuracy requirement of 0.02.

In engineering terms, in this research, the success of the ASM was not intrinsic [28], but was a combination of many factors, including full coverage of the potential degradation mechanisms through a priori knowledge, flexible and robust root models that obtained via offline learning, and sufficient online learning against reliable real-time measurement. Using the known data, one should fine-tune these offline/online training parameters and validate the accuracy of the ASM to improve its applicability.

**Funding:** This research is partially supported by the "Applying the real-time machine learning for the AI reliability modeling of the light output depreciation and color shifting of the micro LED array packaging" project of Feng Chia University, sponsored by the National Science and Technology Council, under the grant no. MOST 111-2221-E-035-043.

**Data Availability Statement:** The data presented in this study are available in this article.

**Conflicts of Interest:** The author declares no conflict of interest.

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
