# Peer review of "An AI-Based Adaptive Surrogate Modeling Method for the In-Service Response of UVLED Modules"

_electronics, doi:10.3390/electronics11182861_

Round 1

Reviewer 1 Report

My biggest issue Is that the title doesn't match the article. There is no connection to digital twins (DTs) in the article. DT is in the abstract and keywords three times. DT is mentioned in the actual article only twice and not in any substantive way. DT isn't referenced and described in the Introduction, which doubles as a literature review. The article itself really doesn't contain any aspects of DTs. The authors need to understand what DTs are and incorporate the concept in their article or remove any reference to DTs.

As referenced above, the Introduction and Literature Review should be two distinct sections.

Line 193 - "A clear and stable deviation in a well-controlled environment indicated that the in- services uncertainty might exist with high probability." This should be elaborated on. If it's uncertainty, that can cover a very wide area. Is it uncertainty or just variance.

Line 168, 261, 306 - "Oracle's point of view". What does this mean?

Line 366 - Is "generic" suppose that or "genetic"?

The Conclusion section needs to be improved. Does the hybrid learning method improve on offline/online machine learning? What about the root models?

Reviewer 2 Report

The study introduces a hybrid adaptive surrogate model for in-service predictions of UVLED module.

Though the work is interesting, in its current form it is not publishable due to a few minor flaws as listed below. The authors are requested to address these comments before the manuscript can be considered for publication.

Give details of the GA model used (convergence diagram), why was the model used instead of more advanced NSGA and LIPO algorithms?

How is the ANN structure initially chosen?

Since being a single output model with a relatively simple structure, why was ANN chosen for this study? Wouldn't a less complex ML model be sufficient for the prediction?

Comment on the interpretability of the ML model.

Round 2

Reviewer 1 Report

The authors addressed my issues, except for one thing. They reference an "oracle's point of view."  They did describe what the point of view is, but don't describe what they mean by an "oracle". Is this an entity with a perfect view of the future or something else? It's an unusual phrase to see in a paper. It needs to be explained.
